# Blockade of Platelet CysLT1R Receptor with Zafirlukast Counteracts Platelet Protumoral Action and Prevents Breast Cancer Metastasis to Bone and Lung

**DOI:** 10.3390/ijms232012221

**Published:** 2022-10-13

**Authors:** Lou Saier, Johnny Ribeiro, Thomas Daunizeau, Audrey Houssin, Gabriel Ichim, Caroline Barette, Lamia Bouazza, Olivier Peyruchaud

**Affiliations:** 1INSERM, Unit 1033, LYOS, Université Claude Bernard Lyon 1, 69372 Lyon, France; 2Cancer Research Center of Lyon (CRCL), INSERM, Unit 1052, CNRS, UMR, 5286 Lyon, France; 3CEA, INSERM, BGE, Université Grenoble Alpes, 38000 Grenoble, France

**Keywords:** platelet, metastasis, breast cancer, cysteinyl leukotrienes, CysLT1R, zafirlukast, LTRA, MGST

## Abstract

Metastases are the main cause of death in cancer patients, and platelets are largely known for their contribution in cancer progression. However, targeting platelets is highly challenging given their paramount function in hemostasis. Using a high-throughput screening and platelet-induced breast tumor cell survival (PITCS) assay as endpoint, we identified the widely used anti-asthmatic drugs and cysteinyl leukotriene receptor 1 (CysLT1R) antagonists, zafirlukast and montelukast, as new specific blockers of platelet protumoral action. Here, we show that human MDA-B02 breast cancer cells produce CysLT through mechanisms involving microsomal glutathione-S-transferase 1/2/3 (MGST1/2/3) and that can modulate cancer cell–platelet interactions via platelet–CysLT1R. CysLT1R blockade with zafirlukast decreased platelet aggregation and adhesion on cancer cells and inhibited PITCS, migration, and invasion in vitro. Zafirlukast significantly reduced, by 90%, MDA-B02 cell dissemination to bone in nude mice and reduced by 88% 4T1 spontaneous lung metastasis formation without affecting primary tumor growth. Combined treatment of zafirlukast plus paclitaxel totally inhibited metastasis of 4T1 cells to the lungs. Altogether, our results reveal a novel pathway mediating the crosstalk between cancer cells and platelets and indicate that platelet CysLT1R represents a novel therapeutic target to prevent metastasis without affecting hemostasis.

## 1. Introduction

Metastases are the leading cause of death in cancer patients and represent a major challenge, as the prognosis remains unfavorable once metastatic dissemination has occurred [1]. There is strong evidence that platelets are involved in tumor cell survival and dissemination. The interactions of circulating tumor cells (CTC) with platelets facilitate metastasis because of both physical interactions and bidirectional activation. The formation of tumor-cell-induced platelet aggregates (TCIPA) facilitate immune evasion and promote vascular adhesion and invasion of tumor cells [2]. Tumor-educated platelets release soluble mediators, thereby enhancing tumor-cell invasiveness and bone metastasis in mice [3,4]. Platelet activation and aggregation have been identified as potential drug targets for cancer therapy. In various models, pharmacological or genetic inhibition of platelets strongly inhibits metastasis [5,6,7]. Despite the well-established role of platelet–tumor-cell interactions in metastasis, the mechanism by which tumor cells educate platelets to promote metastasis remains unclear.

Cysteinyl leukotrienes (CysLTs) LTC_4_, LTD_4_, and LTE_4_ are inflammatory lipid mediators that are key players in many inflammatory processes including tumor development and metastasis. CysLTs are part of the leukotrienes family and are generated from arachidonic acid through the 5-lipoxygenase (5-LOX) pathway to form the unstable LTA_4_ leukotrienes precursor. LTA_4_ is further conjugated to reduced glutathione via leukotriene C_4_ synthase (LTC_4_S) forming LTC_4_ [8]. After export, LTC_4_ is converted to LTD_4_ and then LTE_4_ [9,10]. LTC_4_ and LTD_4_ can bind to two major receptors, CysLT1R and CysLT2R [11,12], whereas LTE_4_ binds to GPR99 [13]. LTC4S is overexpressed in patients with chronic myeloid leukemia and aggressive prostate cancers [14,15]. High expression of CysLT1R is associated with poor prognosis in breast cancer patients [16]. CysLT antagonist (LTRAs) such as montelukast and zafirlukast, which are widely prescribed anti-asthmatic drugs, have been reported to protect asthma patients from developing cancers [17,18]. LTRAs also demonstrated chemopreventive effects in preclinical models [19,20,21,22]. CysLT may modulate the initiation, progression, and metastasis of tumors by directly regulating the proliferation, apoptosis, migration, and invasion of cancer cells as we summarized previously [23]. However, how CysLT contribute to the crosstalk between cancer cells and the inflammatory microenvironment remains poorly understood. Although, LTC_4_ has been shown to activate mice platelets in vivo [24] whether CysLT influence platelet functions to promote cancer progression is unknown.

In the present study, using a high-throughput assay, we identified the LTRAs montelukast and zafirlukast as specific inhibitors of platelet protumoral functions. Our findings demonstrate that CysLT can mediate tumor-cell–platelet interactions and platelet-induced breast cancer progression via platelet CysLT1R. Pharmacological blockade of CysLT1R with zafirlukast prevented bones and lungs colonization of breast cancer cells but did not affect primary tumor growth. Our data reveals a novel platelet-specific signaling pathway for CysLT-induced breast cancer progression. We demonstrate that CysLT1R represents a novel therapeutic target to prevent metastasis, and strongly support the clinical evaluation of LTRAs in combination with anticancer treatments to prevent metastasis in breast cancer patients.

## 2. Results

### 2.1. High-Throughput Screening Identifies CysLT1R Antagonists as Specific Inhibitors of Human Platelet-Induced Human Breast Tumor Cell Survival (PITCS)

In order to identify new anti-metastasis therapies exhibiting a high specificity for the pathological tumor-cell–platelet interactions without abrogating normal platelet functions, we developed a high-throughput screening assay (Figure 1A). Using the Prestwick chemical library of 1280 FDA-approved drugs, we identified that the pharmacological blockade of CysLT1R with zafirlukast and montelukast inhibited the mitogenic action of human platelets on human MDA-B02-Luc breast cancer cells. Both drugs had no impact on cell survival in presence of serum suggesting a specific alteration of platelet protumoral function (Δ activity with platelets vs. serum of 106% for montelukast and 133% for zafirlukast) (Figure 1B). Zafirlukast presented the highest specificity for platelet-induced cells survival, so we then assessed the dose–response of this drug on MDA-B02-Luc cells survival induced by platelets (PITCS) or by serum (Figure 1C). Zafirlukast specifically inhibited the survival of MDA-B02-Luc cells in presence of platelets (IC_50_ = 10 µM) but had no effect in presence of serum even at the higher concentrations. In comparison, everolimus, an mTOR inhibitor that directly targets cancer cells, inhibited MDA-B02-Luc cell survival in both conditions at low micromolar concentrations. Zafirlukast is widely used for the treatment of asthma and no direct alteration of coagulation has been described so far. We found that zafirlukast did not affect TRAP6-induced platelet aggregation in vitro and did not affect coagulation in mice (Appendix A), confirming that zafirlukast does not affect platelet physiological functions and could be used to prevent platelet-induced cancer progression both in mouse models and in clinical practice.

### 2.2. Cysteinyl Leukotrienes Promote Platelet-Induced Survival of Human Breast Cancer Cells via Platelet–CysLT1 Receptor

Platelets are known to not possess 5-LOX and thus do not produce CysLT de novo [25], so we examined whether MDA-B02 cancer cells could be a source of CysLTs. Using a specific CysLT ELISA assay, we confirmed the presence of CysLT (LTC4, LTD4, LTE4) in the conditioned media of both human MDA-B02 and MDA-B02-Luc breast cancer cell lines (Figure 2A). This result confirmed that our breast cancer cells secrete CysLT. As expected, these cancer cell lines express 5-LOX protein, which catalyzes the production of leukotrienes including CysLT, as judged by western blotting (Figure 2B). Surprisingly, LTC4 synthase (LTC4S) was not detected in these cells, whereas it could be found as expected in human platelets and peripheral blood mononuclear cells (PBMC) lysates. LTC4S mRNA was also not detected by RT-qPCR (Appendix A). This result raised the question of how our cancer cell lines are able to produce CysLTs in absence of LTC4S. Non-canonical pathways have been described for CysLT synthesis. For instance, LTC_4_ is efficiently produced by microsomal glutathione-S-transferases MGST2/MGST3 in LTC4S-deficient cells [26,27]. RT-qPCR experiments confirmed that our breast cancer cell lines express all three MGSTs (MGST1/MGST2/MGST3) at steady states (Figure 2C). In order to evaluate the contribution of MGSTs in CysLT secretion by our tumor cells, we first transfected MDA-B02 cells with a series of siRNAs targeting the gene of key enzymes and measured the levels of CysLTs in transfected cell culture media by ELISA (Figure 2D). We found that cells transfected with ALOX-5, MGST1, and MGST2 siRNAs secreted significantly reduced levels of CysLT, whereas transfected cells with MGST3 siRNA only showed a tendency to reduced CysLT secretion. Interestingly, cells with reduced LTA4H expression after siRNA transfection revealed a significantly higher secretion of CysLTs (Figure 2D). This result indicates that inhibition of the LTB4 leukotriene metabolic pathway via LTA4H in these cells stimulated CysLT production suggesting that arachidonic acid metabolism may be switched from one pathway to another. It is well-known that inhibition of specific enzymes of the arachidonic pathway can lead to the elevation of lipid mediators due to pathway shunting. Such metabolic pathway switching in eicosanoid family has previously been described in vitro and in vivo as the absence of LTA4H can lead to the elevation of CysLT [28]. The involvement of MGSTs in CysLT secretion by MDA-B02 cell was further confirmed in the culture media of cells incubated in presence of arachidonic acid (AA), the precursor of all eicosanoids, and treated with A23187, a calcium ionophore well-known to promote CysLT synthesis (Figure 2E). In addition, cell treatment with MK886, a FLAP inhibitor, also inhibited the secretion of CysLTs in presence of AA and A23187. These results further confirmed the high potency of the 5-LOX pathway in our cells.

Given that LTD4/CysLT1R signaling has been described to directly promote cancer cell proliferation and survival, we sought to determine whether MDA-B02-Luc cells might directly respond to CysLT (Figure 3A). Surprisingly, we observed that the stimulation of MDA-B02-Luc cells with LTD4 did not increase cancer cell survival. In addition, LTD4 did not increase PITCS, which might be due to already reached maximum cell survival in presence of platelets. All CysLT, including LTC_4_ and LTD_4_, can bind both CysLT1R and CysLT2R receptors. To assess the specificity of CysLT1R in PITCS, co-cultures of breast cancer cells with platelets were treated with HAMI3379, a CysLT2R inhibitor, or with the CysLT1R antagonist zafirlukast. The inhibition of PITCS was only observed with CysLT1R blockade (Figure 3B). Human platelets express both CysLT1R and CysLT2R, as it is a common feature of many hematopoietic cells [29]. Using qRT-PCR, we were unable to detect the expression of CysLT1R in our two MDA-B02 breast cancer cell lines whereas CysLT1R expression was detected in human platelets from three different healthy donors (Figure 3C). Combined to our previous findings on zafirlukast effect on PITCS, these results identified for the first time that breast-cancer-cell-derived CysLT acts on platelet CysLT1R to mediate PITCS. These results also identified MDA-B02 cells as ideal tools to investigate the paracrine action of cancer-cell-derived CysLT on the tumoral microenvironment and, in this study, more specifically on platelet prometastatic action.

### 2.3. Cysteinyl Leukotrienes Mediate Tumor-Cell-Induced Platelet Aggregation

Then, we assessed the putative role of CysLT in tumor-cell-induced platelet aggregation (TCIPA), a process that contribute to platelet protumoral functions. MDA-B02 cells were incubated for 48 h with an excess of human platelets and platelet aggregates were analyzed on live cells by holographic microscopy. Remarkably, MDA-B02 cells enhanced the formation of platelet aggregates (~10 aggregates) compared to platelets incubated alone (~2 aggregates). TCIPA was significantly reduced by 70% with zafirlukast at 10 µM (Figure 4A). Then, we investigated the effect of CysLT1R blockade on platelet adhesion using confocal microscopy (Figure 4B). The degree of platelet adhesion or “cloaking” was measured based on the fluorescent detection of labelled platelets on the surface of cancer cells post-incubation using anti-CD41 antibodies. The percentage of MDA-B02 cells cloaked with platelets was significantly reduced with zafirlukast (−50% at 1 µM and −80% at 10 µM). It is known that endogenous ADP can amplify platelet activation through P2Y_1_ and P2Y_12_ receptors [30] and P2Y_12_ receptors have been implicated in cellular responses to CysLT [31]. We found that the ability of platelets to aggregate and adhere to cancer cells was also reduced in the presence of clopidogrel, a P2Y_12_ inhibitor, suggesting that the CysLT1R-dependent pathway of TCIPA potentially requires an autocrine ADP-mediated response through P2Y_12_ receptors.

### 2.4. Cysteinyl Leukotrienes Promote Platelet-Induced Breast Cancer Cells Migration and Invasion

Platelets provide key signals that affect tumor cell invasiveness and metastatic potential [32]. To characterize the functional impact of CysLT1R-dependent platelet–cancer-cell interaction, we investigated the effect of platelets on the invasive behavior of MDA-B02 cells. In a wound-healing assay, incubation with platelets for 5 h significantly stimulated the migration of MDA-B02 cells to levels similar to the migration induced by serum (Figure 5A). Wound closure and cell migration in presence of platelets were increased by a factor of 3.2 and 3.6, respectively. This effect was blocked by zafirlukast at 10 µM, whereas this drug did not affect serum-induced migration, excluding the possible off-target effects of zafirlukast in inhibiting platelet-induced migration. In addition, down-regulation of ALOX5, MGST2, and MGST3 genes using specific siRNAs also decreased platelet-induced cancer cell wound closure and cell migration rate (Figure 5B). We next investigated if platelet-cloaked cancer cells had an altered invasive phenotype. MDA-B02 cells were incubated alone or with platelets and allowed to invade in Matrigel^®^ chambers for 48 h. Platelets significantly increased, by 58%, the ability of cancer cells to invade, and zafirlukast did not affect the basal invasion activity of cancer cells (Figure 5C). In contrast, prevention of CysLT1R-dependent platelet aggregation using zafirlukast effectively reduced platelet-cloaked MDA-B02 cell invasion to levels comparable to those of MDA-B02 cells alone. This finding reveals the role of tumoral CysLTs and platelet CysLT1R in mediating platelet-induced migratory and invasive phenotypes in MDA-B02 breast cancer cells.

### 2.5. Blockade of CysLT1R with Zafirlukast Prevents Early Bone Colonization of Human Breast Cancer Cells in BALB/c Nude Mice

In light of the important role of platelets in metastatic dissemination, we then investigated whether CysLT1R blockade with zafirlukast would impact metastasis in vivo. We injected MDA-B02-Luc cells into the tail artery of BALB/c nude mice that received a daily treatment with zafirlukast (0.4 mg/kg per os) (Figure 6A). After 7 days, bone marrow cells were collected and placed in culture in presence of puromycin for 14 days. After staining with crystal violet, clones of tumor cells that colonized bone (TCB) were quantified. Treatment with zafirlukast strongly decreased the incidence of mice presenting bone colonization by cancer cells (three out of eight mice) compared to vehicle (seven out of seven mice) (Figure 6B). Moreover, zafirlulast significantly inhibited by 90% the number of TCB indicating the major role of CysLT1R of the tumor microenvironment in the metastatic process (Figure 6C).

### 2.6. Blockade of CysLT1R with Zafirlukast Prevents Spontaneous Lung Colonization by 4T1 Cells without Affecting Orthotopic Tumor Growth in a Syngeneic Mouse Model

In order to confirm the anti-metastasis property of zafirlukast, we used a different mouse model consisting of syngeneic animal model of spontaneous metastasis dissemination of murine 4T1 breast cancer cells. After orthotopic xenograft in the mammary gland of BALB/c immunocompetent mice, the animals were treated with zafirlukast (0.4 mg/kg/day, i.p.), with the common chemotherapeutic agent paclitaxel (30 mg/kg/2 days, i.p), or with both drugs combined, starting post-tumor cell injections at day 3 until sacrifice at day 12 (Figure 7A). Lungs were collected, minced, treated with collagenase, and placed in culture for 14 days in presence of 6-thioguanine to select for 4T1 tumor cells which are naturally resistant. After staining with crystal violet, clones of tumor cells that colonized the lungs (TCL) were quantified. In these conditions, treatment with zafirlukast or paclitaxel alone strongly decreased the incidence of mice presenting spontaneous lung colonization by cancer cells (3 out of 10 mice or 4 out of 9 mice, respectively) compared to vehicle (10 out of 10 mice) (Figure 7B). Moreover, the number of TCL was reduced by 88% with zafirlukast compared to 60% with paclitaxel (Figure 7C). Remarkably, combined treatment of zafirlukast plus paclitaxel inhibited lung colonization by 100% (Figure 7B,C). These results confirmed the potent inhibitory activity of zafirlukast on breast cancer cell metastatic spreading. Platelets have been shown to contribute to cancer growth [4,33] and our in vitro experiments described previously demontrated that platelets promote breast cancer cell survival. Then, we examined whether zafirlukast could affect tumor growth using the 4T1 model. In contrast to paclitaxel, zafirlukast alone had no significant effect on primary tumor growth (Figure 7D,E). Moreover, in vivo, zafirlukast did not provide any additive anti-tumoral property to paclitaxel (Figure 7D,E), as opposed to what was observed in vitro in PITCS when cells were cotreated with increased concentrations of zafirlukast in combination with paclitaxel (Appendix A). These results suggest that CysLT1R signaling might specifically be involved in the metastatic process.

## 3. Discussion

Platelets contribute to the metastatic dissemination in a variety of ways, including coating tumor cells to help them evade the immune system, shielding tumor cells from high shear forces, and facilitating the adhesion of tumor cells to the vascular endothelium [32]. However, while the ability of cancer cells to aggregate and cluster platelets around CTC correlate with increased metastatic disease, as supported by a large body of experimental and clinical data, the underlying mechanisms are not fully understood [34,35,36,37]. There are many pharmacological agents that are known to block platelet functions and are thought to have anti-metastatic effect. Regular use of aspirin has been shown to decrease the incidence of several cancers, as well as the development of metastasis [38]. However, its long-term use can increase bleeding risk [39]. In light of the strong need for new therapies that could specifically target pathological tumor-cell–platelet interactions without abrogating normal platelet functions, we identified here for the first time CysLT1R antagonists, such as zafirlukast, as specific inhibitors of platelet protumoral functions. Several studies have shown that CysLT contribute to cancer progression by directly promoting cancer cells survival and invasion, mostly via LTD4/CysLT1R signaling [23]. However, whether CysLT contributes to platelet-induced cancer progression and/or metastasis has not yet been addressed.

In the present study, we show that the metastatic human MDA-B02 breast cancer cells secrete CysLT through a mechanism involving MGSTs but do not express CysLT1R. On the other hand, platelets constitutively express CysLTR but cannot produce CysLT de novo [25,29]. This model allowed us to study the effect of tumoral CysLT exclusively on platelet functions, avoiding autocrine stimulations. Previous studies have found that there is a direct correlation between the ability of CTC to interact with platelets via TCIPA and their increased survival in circulation and subsequent formation of metastases [40,41]. The blockade of platelet CysLT1R with zafirlukast significantly attenuated platelets aggregation and adhesion to cancer cells. Platelet aggregation by tumor cells is a complex process involving different pathways. Several cell lines have been shown to induce TCIPA in a P2Y_12_-dependent manner, either by the direct secretion of ADP or by the induction of ADP release by platelets themselves [36,37,42]. We found that platelet aggregation induced by MDA-B02 cells was sensitive to clopidogrel, suggesting that MDA-B02 induces the release of ADP from platelet-dense granules to facilitate the induction of aggregation. A previous study from Cummings and colleagues has indeed shown that LTC_4_ can activate mouse platelets and induce ADP release in a P2Y_12_-dependent manner [43]. Subsequent activation and aggregation of platelets may thus cause a cycle of platelet recruitment and protection of CTC that ultimately facilitates tumor cell survival in the circulation and at metastatic sites. It has been previously described that CysLT does not directly induce platelet aggregation but potentiates the effect of platelet agonist in vitro [44]. Our results suggest that tumoral CysLT potentiate the pro-aggregatory effect of tumor cells on platelets and thus contribute to metastasis. Cummings and colleagues describe that LTC_4_, but not LTD_4_ or LTE_4_, activate mouse platelets exclusively through CysLT2R [43]. In contrast, we showed that CysLT2R blockade with HAMI3379 does not inhibit platelet-induced proliferation of MDA-B02 cancer cells, indicating the major role of CysLT1R in the mediation of CysLT signaling in human platelets. Platelets are important modulators of the capacity of tumor cells to invade and metastasize. We found that platelets had a significant stimulatory effect on the migratory and invasive capacity of MDA-B02 cells, which was abolished with zafirlukast and by inhibition of CysLT secretion by tumor cells. We thus demonstrated that tumor-cell-derived CysLT promotes platelet-induced metastatic potential of breast cancer cells via platelet CysLT1R. This finding is supported by the absence of effect of zafirlukast on cancer-cell-invasive phenotypes in absence of platelets. The ratio of platelets to cancer cells used in in vitro experiments was 30–300 times lower than physiological concentrations, suggesting that levels of active platelets in promoting cancer progression are present in vivo. Finally, it is noteworthy to mention that the micromolar doses of zafirlukast we used in our study are similar to the plasma concentrations in patients following a single 20 mg dose, according to the pharmacokinetic data of Accolate^®^ (zafirlukast).

The discovery of platelet CysLT1R as an important mediator of platelet aggregation and adhesion in response to MDA-B02 cancer cells makes it a strong potential target for therapeutic intervention. We demonstrated that CysLT1R blockade with zafirlukast could prevent breast cancer metastasis to bone and lungs in two different mice models. First, zafirlukast significantly decreased dissemination to bone of human breast cancer cells MDA-B02 in nude mice. MDA-B02 does not express CysLT1R, indicating that CysLT mainly contributes to the metastatic process by acting on the tumor cell microenvironment, rather than on cancer cells themselves. Secondly, zafirlukast inhibited spontaneous dissemination to the lungs of murine breast cancer cells 4T1 in syngeneic mice. Notably, our results support the major role of CysLT1R in the metastatic process, as zafirlukast did not affect primary tumor growth.

The inhibition of the platelet CysLT1R, although not targeting the tumor cell itself, could potentially reduce metastasis and prevent cancer-associated thrombotic events without compromising the hemostatic functions of platelets. In support of this idea, LTRAs have been used for more than two decades for the treatment of asthma and allergic rhinitis [45]. Moreover, a cohort study of 4185 asthma patients showed that the use of LTRAs significantly lowered cancer incidence rate, especially for lung, colorectal, and breast cancer [17]. Recently, a long followed-up study on 188,906 participants confirmed that LTRA use was associated with an overall decreased risk of cancer [18]. We also found that the combination of zafirlukast with paclitaxel show an additive chemopreventive effect both in vitro and in vivo, supporting the clinical evaluation of CysLT1R antagonists in combination with chemotherapies to prevent metastases before they occur.

## 4. Materials and Methods

### 4.1. Cell Culture and Reagents

Two osteotropic metastatic human breast cancer cell lines were used in this study: MDA-MB-231/B02 (MDA-B02) and a subclone expressing the three markers GFP, luciferase, and a beta-galactosidase referred to as MDA-B02-Luc for simplification and which was described previously [46], as well as their parental breast cancer cell line MDA-MB-231 (MDA-231) and the murine 4T1 breast cancer cells obtained from the American Type Culture Collection (ATCC^®^). The 4T1 cells derive from a BALB/c spontaneous mammary carcinoma and are naturally resistant to 6-thioguanine [47]. They were maintained in Dulbecco’s Modified Eagle Medium supplemented with 10% fetal bovine serum (FBS) and 1% penicillin/streptomycin. Zafirlukast and montelukast, the CysLT1R antagonists, everolimus, the mTORC1 inhibitor, and clopidogrel, the P2Y_12_ inhibitor, were purchased from Sigma. HAMI3379, the CysLT2R antagonist, and MK-886, the FLAP inhibitor, were from Cayman. LTD_4_ was from Bertin.

### 4.2. Platelet Preparation

Whole blood was collected by venipuncture of fully informed healthy volunteers. Platelet-rich plasma (PRP) was prepared from trisodium-citrated blood centrifuged at 200× *g* for 15 min. For the preparation of washed platelets, PRP was centrifuged at 1000× *g* for 10 min with 10 mM PGI2 and 10 U/mL apyrase to prevent platelet activation. The platelet pellet was resuspended in tyrode HEPES buffer (NaCl 137 mM; KCl 3 mM; NaHCO_3_ 12 mM; Na_2_HPO_4_ 0.34 mM; MgCl_2_ 1 mM; glucose 5 mM; Hepes 20 mM; pH 7.3).

### 4.3. CysLT Enzyme Immunoassay

Cells were seeded at a density of 5 × 10^5^ in 6-well plates and cultured for 48 h in serum-free media. The cell media were collected and centrifuged at 11,000× *g* for 5 min at 4 °C. Alternatively, cells were incubated for 30 min in PGC buffer (1mM CaCl_2_, 1mg/mL glucose) with 40 µM arachidonic acid (VWR) and 5 µM A23187 calcium ionophore (Sigma) to stimulate CysLT production. Secretion of CysLT in the cell culture media was measured with an enzyme immunoassay performed according to the manufacturer’s instructions (Cayman Chemical Company, Ann Arbor, MI, USA). Briefly, a standard curve (r^2^ = 0.99) was calculated from a series of wells containing a known amount of analyte on the same plate as the samples to validate the assay. The concentrations indicated were corrected to take into account the background noise caused by the culture media, according to the manufacturer instructions (Cayman #500390).

### 4.4. Western Blot

Total proteins were harvested with RIPA lysis buffer (Sigma Aldrich, St. Louis, MO, USA) containing a protease inhibitor cocktail (cOmplete™, Roche^®^, Sigma Aldrich, St. Louis, MO, USA). Total proteins lysates (30 µg) were separated on a 4–20% Mini-PROTEAN^®^ TGX™ Precast Stain-free Protein Gel (Biorad, Hercules, CA, USA) and transferred in 5 min at 20 V on a nitrocellulose membrane using the iBlot™ Transfer Device (ThermoFisher, Waltham, MA, USA). Membranes were probed with antibodies against 5-LOX (PA5-95034, Invitrogen, 1:2000) and LTC4S (TA330969, Origene, 1:500). Western blot signals were detected with a chemiluminescence assay (Clarity Western ECL, Biorad, Hercules, CA, USA). Total proteins were visualized using the stain-free technology.

### 4.5. Quantitative RT-PCR

Total RNA extraction was performed on TRIzol lysed cells with the NucleoSpin RNA Plus kit (Macherey-Nagel), and cDNA synthesis was performed with the iScript kit (Biorad), according to the manufacturers’ instructions. Expression of the targets was quantified by real-time quantitative RT-PCR (qRT-PCR) using gene-specific primers: CYSLT1R: qHsaCID0008825; CYSLT2R: qHsaCED0019603; MGST1: qHsaCID0006578; MGST2: qHsaCED0057437; MGST3: qHsaCID0018341; LTC4S: qHsaCID0020567 (PrimePCR™ Primers, Biorad). The data were normalized to the expression of the RPL32 gene (ribosomal protein L32).

### 4.6. siRNA Transfection 

Cells were seeded at a density of 3 × 10^5^ in a 6-well plate in DMEM supplemented with 10% FBS and transfected the next day using Silencer^®^ Select siRNA: ALOX5: s1275; LTA4H: s8307; MGST1: s8757; MGST2: s8759 + s8760; MGST3: s8762 (Ambion™); and Lipofectamine^®^ 2000, according to the manufacturer’s instructions. Cells were incubated for 2–4 days before the next experiment.

### 4.7. Cell Survival Assay

MDA-B02-Luc cells were seeded at a density of 1.5 × 10^3^ in a 384-well plate with serum-free F-12 media. The next day, the wells were supplemented with 5 × 10^4^ human platelets (ratio of 333:1 tumor cell) or 10% FBS. The cells were incubated for 48 h before measuring the bioluminescence with the ONE-Glo™ Luciferase assay (Promega). High-throughput screening of the Prestwick chemical library (1280 compounds) was performed in duplicate and in presence of platelets or FBS as described above, using the robotic facility from the CMBA platform (CEA Grenoble). For the automated assay protocol, several integrated instruments were used, including the Tecan’s MCA96 96-channel pipetting head and the Tecan’s Infinite M1000 microplate reader. Drugs were applied to cell cultures just before supplementation with platelets or FBS. Drugs were used at 10 µM for primary screening. In each test plate, 32 positive control points (in presence of FBS) and 32 negative control points (without FBS) were included, and data values were used to calculate the Z’ factor [48] in order to monitor the assay quality. The % ∆ activity (platelets vs. FBS) was calculated from the difference between the platelet activity inhibition and the FBS activity inhibition of the drug. All drugs that inhibited platelet-induced cancer cell survival without effecting serum-induced cell survival were subjected to a large dose–response assay as a secondary screening test.

### 4.8. Live-Cell Imaging of Tumor-Cell-Induced Platelet Aggregation

Cells were seeded at a density of 2 × 10^5^ in 35 mm µ-Dish (Ibidi) for 16 h in DMEM with 10% FBS. The next day, the media was replaced with serum-free F-12 and 75 × 10^6^ human platelets was added (ratio of 340:1 tumor cell). The cells were then treated with zafirlukast, clopidogrel, or MK-886 and incubated for 48 h at 37 °C. Platelets aggregates were examined under a holographic microscope (3D live cell imaging, Nanolive), and four images per well were captured to quantify aggregates (>3 platelets).

### 4.9. Platelet Adhesion Assay

Cells were seeded at a density of 6 × 10^4^ in 4-well Nunc Lab Tek chamber slides for 16 h in DMEM with 10% FBS. The next day, the media was replaced with serum-free F-12 and 20 × 10^6^ human platelets were added (ratio of 333:1 tumor cell). The cells were then treated with zafirlukast, clopidogrel, or MK886 and incubated for 48 h at 37 °C. Next, the cells were washed with PBS and fixed with 4% paraformaldehyde diluted in PBS for 15 min. The cells were permeabilized with 0.2% Triton X-100 in PBS for 5 min and blocked with 1% BSA for 1h. The cells were incubated with the A488-conjugated phalloidin probe and the anti-CD41-PE antibody overnight at 4 °C in the dark. Cells were counterstained with DAPI for the nucleus. The cells were examined under a Zeiss 880 fluorescence microscope, and four images per well were captured to quantify platelet aggregates bound to cancer cells. The experiment was performed three times.

### 4.10. Scratch Assay

MDA-B02 were seeded at a density of 1 × 10^6^ in a 6-well plate for 16 h in DMEM with 10% FBS. The cell monolayer was scraped in a straight line to create a “scratch” using a P200 pipet tip. Subsequently, the cells were washed twice with phosphate-buffered saline (PBS). The cells were then incubated with 50 × 10^6^ human platelets (ratio of 50:1 tumor cell) in serum-free F-12 or without platelets in F-12 with 10% FBS. Photographs of the treated cells migrating within the scratch were captured at 0 and 5 h. Three independent experiments in triplicate were performed.

### 4.11. Invasion Assay

Cell invasion assay in vitro was performed using 6.5 mm Transwell chambers with 8 µm pores (Corning) coated with growth-factor-reduced Matrigel (Corning, 0.3 mg/mL diluted with serum-free media). MDA-B02 cells were seeded at a density of 50 × 10^4^ in the upper chamber in F-12 with 0.1% BSA. 17 × 10^6^ human platelets (ratio of 340:1 tumor cell) and zafirlukast was added in selected wells. F-12 with 1% FBS was used as a chemoattractant in the lower chamber or F-12 with 0.1% BSA for the negative control wells. Cell invasion was allowed to occur for 48 h, and cells on the top membrane surface were removed with cotton swabs. The invading cells were stained with DAPI and counted from three randomly selected microscopic fields. Three independent experiments in triplicate were performed with human platelets from three different healthy volunteers.

### 4.12. Early Bone Colonization of Breast Cancer Cells in Mice

MDA-B02-Luc cells (5 × 10^5^ in phosphate-buffered saline) were injected into the tail artery of 5-week-old female BALB/c nude mice as previously described [4]. Animals were treated per os with zafirlukast according to the doses used with humans (0.4 mg/kg/day) or with placebo (physiological serum). Seven days post-injection, the mice were euthanized and the hind limbs were dissected. Bones were chopped and treated with 0.25 mg/mL collagenase for 2 h at 37 °C. The cell suspension was washed with phosphate-buffered saline and resuspended in complete media supplemented with 1 mg/mL puromycin. After 2 weeks, the clones were fixed and stained with a solution of crystal violet and counted.

### 4.13. Syngeneic Model of Lung Metastasis

The 4T1 cells (1 × 10^5^ in phosphate-buffered saline) were injected in the fat pad of the mammary gland of 6-week-old female BALB/c mice (Janvier Labs) as previously described [49]. Animals were treated from day 3 with zafirlukast according to the doses used with humans (0.4 mg/kg/day), paclitaxel (30 mg/kg/2 days), or a combination of both for twelve days. Then, the mice were euthanized, primary tumors were surgically removed and weighed, and lungs were collected and treated with 0.25 mg/mL collagenase. The lung cell suspension was resuspended in complete media supplemented with 6-thioguanine (10 µg/mL) to select the resistant 4T1 cells. After 2 weeks, the clones were fixed and stained with a solution of crystal violet and counted.

### 4.14. Statistical Analyses

Differences between groups were determined by 1-way or 2-way analysis of variance (ANOVA), followed by a Bonferroni posttest using GraphPad Prism version 8.4.3 software. Single comparisons were carried out using the non-parametric Mann–Whitney test; *p* < 0.05 was considered statistically significant.

## 5. Conclusions

In conclusion, our findings reveal a novel pathway mediating the crosstalk between cancer cells and platelets. We describe for the first time that blocking the platelet–CysLT1R receptor altered the protumoral action of blood platelets and prevented metastasis of breast cancer cells to bone and lungs. Because LTRA such as zafirlukast and montelukast are commonly used clinically for the treatment of asthma, our data suggest that they could be promising repurposing therapeutic options for preventing tumor metastasis.

## Figures and Tables

**Figure 1 ijms-23-12221-f001:**
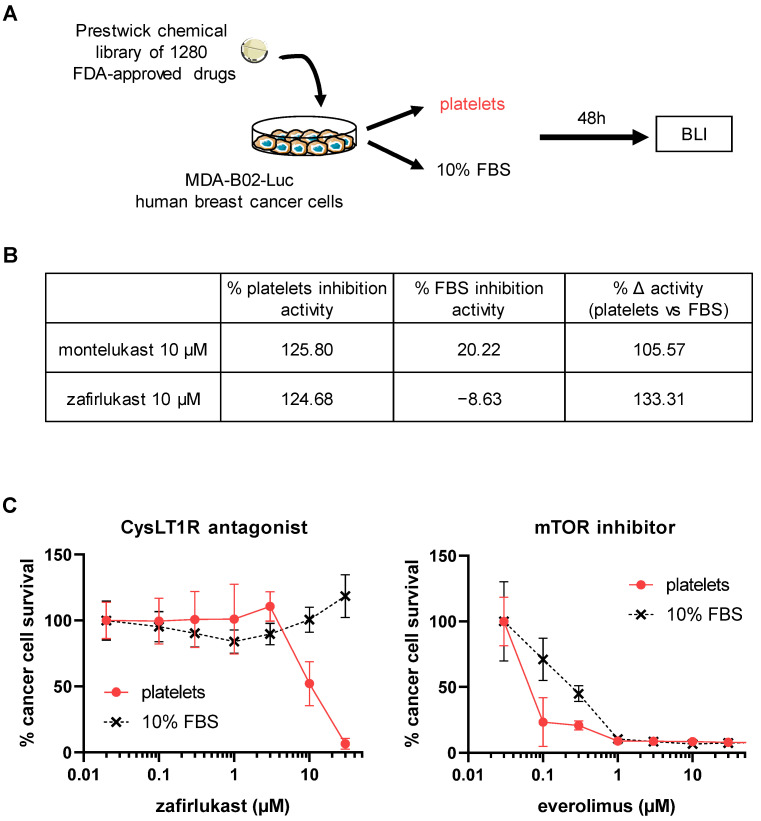
High-throughput screening identifies CysLT1R antagonists as specific inhibitors of human platelet-induced survival of human breast cancer cells. (**A**) A high-throughput screening using the Prestwick chemical library of 1280 FDA-approved drugs has been developed to identify compounds that inhibit platelet protumoral functions. For this, MDA-B02-Luc breast cancer cells were cultured for 48 h with human platelets in serum-free media, or with media supplemented with 10% FBS. The survival of cancer cells was measured using a bioluminescence assay (BLI). The % ∆ activity (platelets vs. FBS) was calculated from the difference between the platelet activity inhibition and the FBS activity inhibition of the drug. (**B**) CysLT1R antagonist, montelukast, and zafirlukast showed a high specificity for the inhibition of platelet-induced cancer cell survival at 10 µM (% Δ activity). (**C**) Dose–response curve of the CysLT1R antagonist zafirlukast and the mTOR kinase inhibitor everolimus on platelet- and serum-induced cancer cell survival. Data is the mean ± SD and is representative of three independent experiments. Abbreviations: BLI = bioluminescence; FBS = fetal bovine serum.

**Figure 2 ijms-23-12221-f002:**
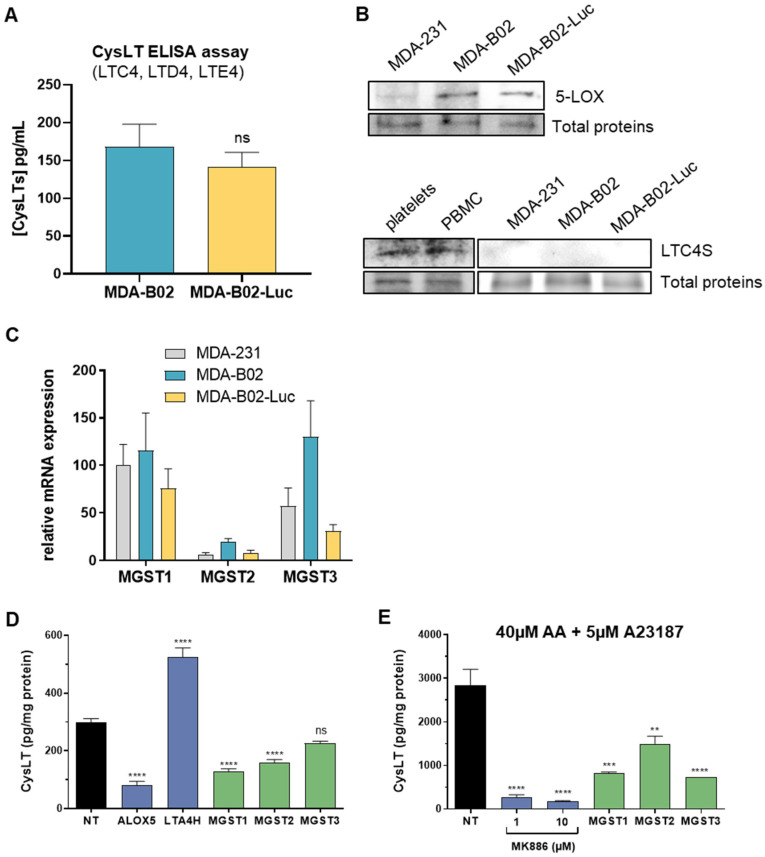
MDA-B02 breast cancer cells produce CysLT via MGST enzymes in absence of LTC4S. (**A**) The presence of secreted CysLT (LTC4, LTD4, LTE4) in the conditioned media of MDA-B02 and MDA-B02-Luc cells was analyzed using an ELISA assay. (**B**) 5-Lipoxygenase (5-LOX) and LTC4 synthase (LTC4S) were analyzed in the two MDA-B02 cell lines and in their parental MDA-231 cell line using western blotting assays. (**C**). RT-qPCR analysis of the three microsomal glutathione S-transferase enzymes (MGST1/2/3). (**D**) CysLT secreted in the conditioned media of MDA-B02 cells transfected with siRNA targeting the leukotriene pathway or MGST1/2/3 were analyzed by ELISA assay. (**E**) MDA-B02 cells treated with the leukotriene biosynthesis inhibitor MK886 or transfected with siRNA targeting MGST1/2/3 were stimulated with 40 µM arachidonic acid (AA) and 5 µM calcium ionophore A23187 for 30 min before analyzing the secreted CysLT using ELISA assay. (ns = not significant; ** *p* < 0.05; *** *p* < 0.001; **** *p* < 0.0001; vs. NT; one-way ANOVA). Abbreviations: AA = arachidonic acid; CysLT = cysteinyl leukotriene; PBMC = peripheral blood mononuclear cell.

**Figure 3 ijms-23-12221-f003:**
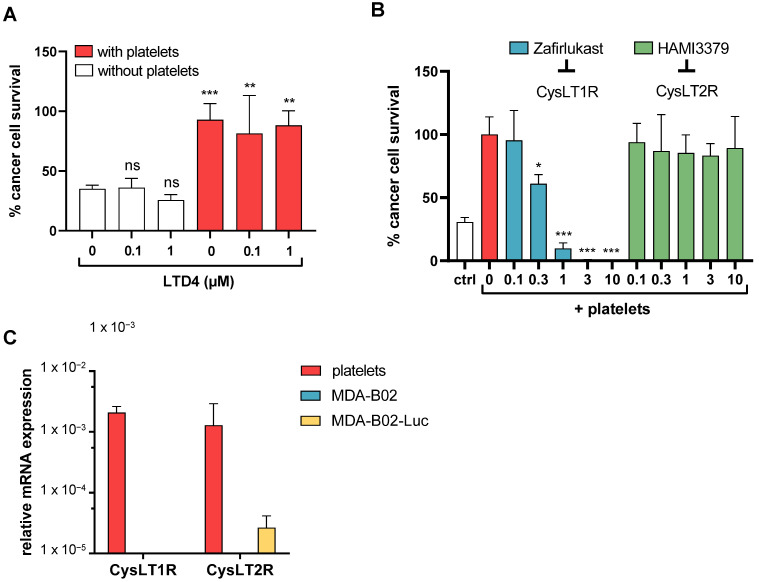
Cysteinyl leukotrienes promoting platelet-induced survival of human breast cancer cells exclusively via platelet–CysLT1 receptor. (**A**) Effect of LTD4 on MDA-B02-Luc cell survival in absence or presence of platelets (ns = not significant; ** *p* < 0.005; *** *p* < 0.001; vs. NT without platelets; one-way ANOVA). (**B**) Effect of zafirlukast, a CysLT1R antagonist, or HAMI 3379, a CysLT2R antagonist, on platelet-induced MDA-B02-Luc cell survival. All concentrations are in µM (* *p* < 0.05; *** *p* < 0.001; vs. NT; one-way ANOVA). (**C**) RT-qPCR analysis of CysLT receptors in human platelets or MDA-B02 cancer cells. Human platelets come from three different donors.

**Figure 4 ijms-23-12221-f004:**
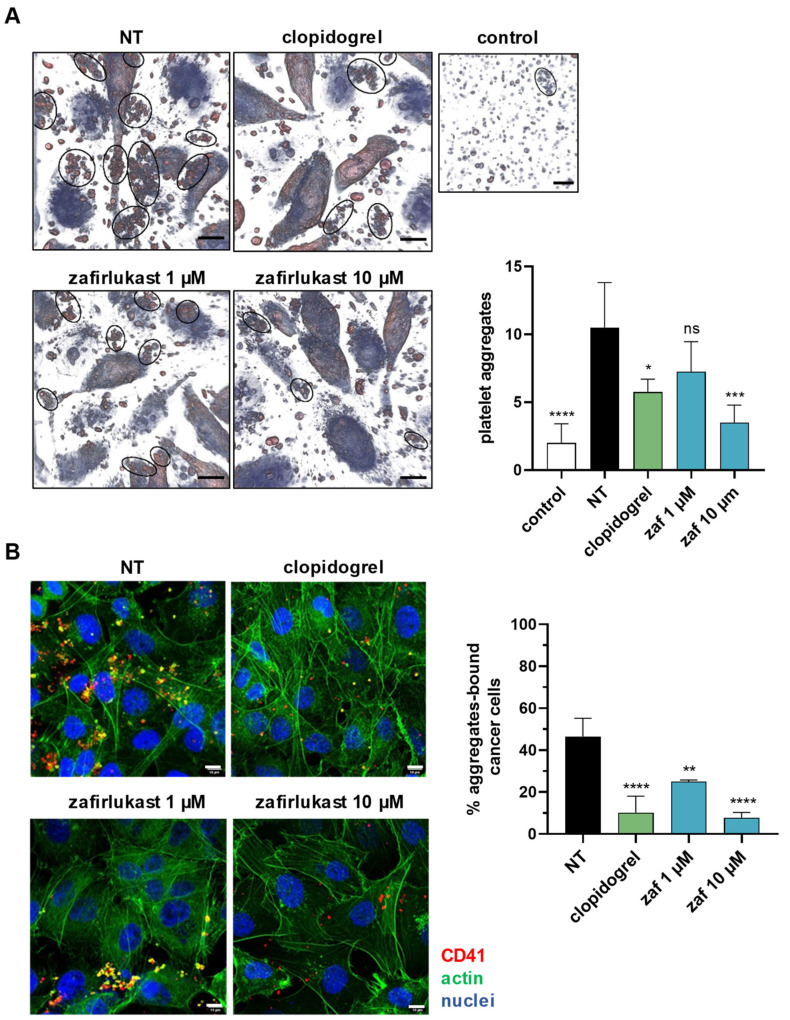
Cysteinyl leukotrienes mediate tumor-cell-induced platelet aggregation (TCIPA). (**A**) MDA-B02 breast cancer cells were cultured with human platelets and the different drugs for 48 h. Platelet aggregates (black circles) were imaged using a 3D Live Cell Explorer (Nanolive) and quantified. The data represents the mean ±SD of four images. Scale: 100 µm. (**B**) After 48 h of culture with MDA-B02 cells, non-adherent platelets were washed. Platelet aggregates bound to cancer cells were stained using an anti-CD41 antibody (red) and quantified using confocal microscopy. Actin from cells was stained using a A488-conjugated phalloidin probe (green) and nuclei were stained with DAPI (blue). The data represent the mean ± SD (n = 3), and the platelets come from two different donors. Scale: 10 µm (ns = not significant; * *p* < 0.05; ** *p* < 0.005; *** *p* < 0.001; **** *p* < 0.00001 vs. NT; one-way ANOVA). Abbreviations: NT = non-treated; zaf = zafirlukast.

**Figure 5 ijms-23-12221-f005:**
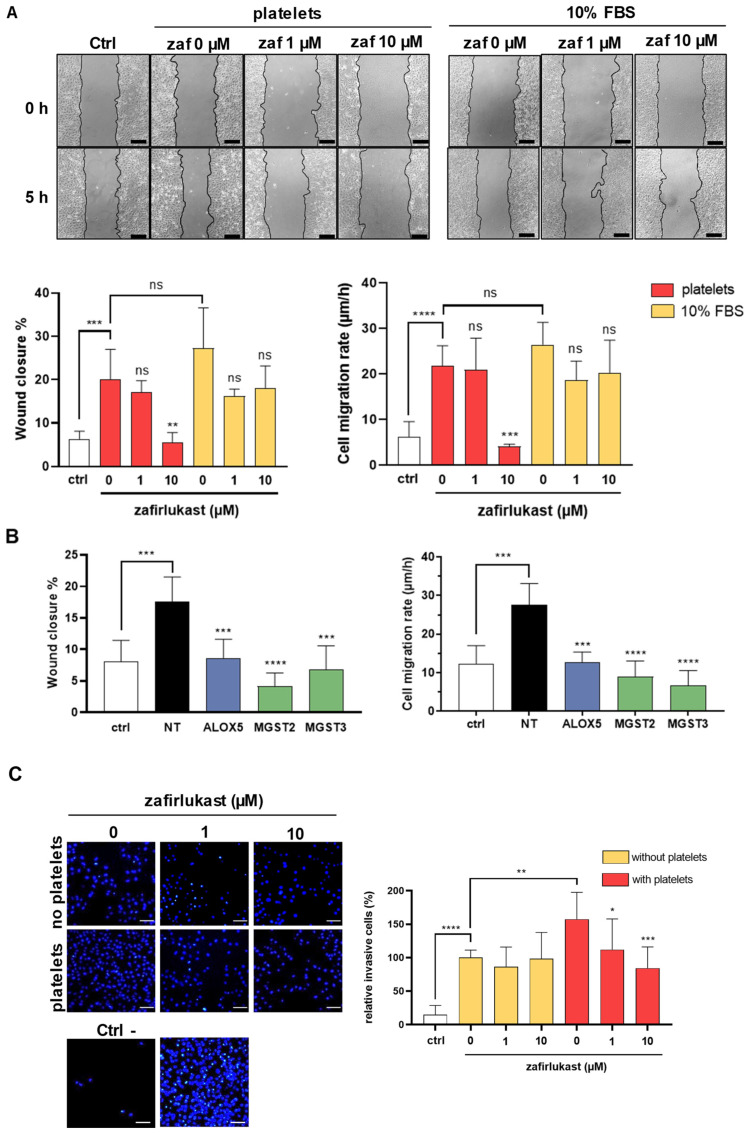
Platelet CysLT1R contribute to platelet-induced MDA-B02 breast cancer cell invasive phenotypes. (**A**) Representative bright-field images of a wound healing assay showing that zafirlukast at 10 µM prevents platelet-induced migration of MDA-B02 cells but not serum-induced migration. Scale: 100 µm. Wound closure (%) is expressed as the remaining area uncovered by the cells at h = 5. Cell migration rate (µm/h) is calculated from the measure of wound width. The data represent the mean ± SD (n = 3) and the platelets come from two different donors. The control group (ctrl) is cultivated in absence of platelet and FBS. (**B**) The inhibition of tumoral CysLT production by MDA-B02 cells transfected with siRNA targeting ALOX5, MGST2, or MGST3 inhibited platelet-induced cancer cell migration as measured by the decrease of wound closure and cell migration rate (µm/h) after 5 h of co-culture in presence of human platelets. The control group (ctrl) was cultivated in absence of platelets and FBS. NT = Non-transfected cells. (**C**) Effect of zafirlukast on the platelet-induced invasion activity of MDA-B02 cells. Cells were cultured in 0.1% BSA and allowed to invade Matrigel^®^ chambers for 48 h in presence or absence of platelets and zafirlukast. Media supplemented with 1% FBS was used as a chemoattractant in the lower chamber, with 0.1% BSA for the negative control wells (ctrl -), or with 10% SVF for the positive control wells (ctrl +). The data represent the mean ±SD (n = 3), and the platelets comes from three different donors. Scale: 100 µm (ns = not significant; * *p* < 0.05; ** *p* < 0.005; *** *p* < 0.001; **** *p* < 0.00001 vs. NT; one-way ANOVA). Abbreviations: ctrl = control, FBS = fetal bovine serum, NT = non-tranfected.

**Figure 6 ijms-23-12221-f006:**
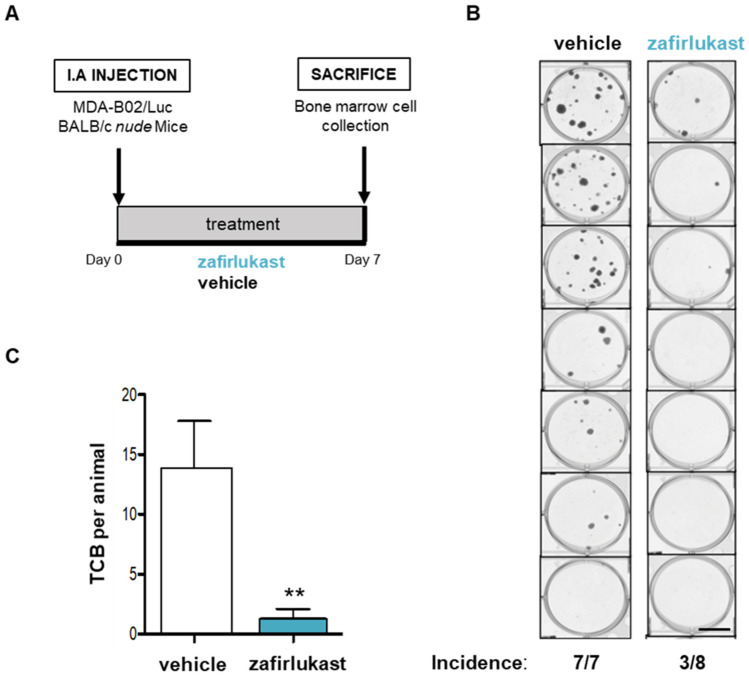
Blockade of CysLT1R with zafirlukast prevents early bone colonization of human breast cancer cells in BALB/c nude mice. (**A**) MDA-B02-Luc cells were injected into the tail artery (IA) of BALB/c nude mice that received a daily treatment with zafirlukast (0.4 mg/kg per os) or the vehicle (physiological serum). After 7 days, bone marrow cells were collected and placed in culture in presence of puromycin (1 µg/mL) for 14 days. (**B**) Tumor cells that colonized the bone (TCB) were fixed and stained by crystal violet (scale: 100 µm). (**C**) TCB were quantified. The data represent the mean number of TCB ± SD (vehicle: 7 mice; zafirlukast: 8 mice). ** *p* < 0.005 vs. vehicle; Mann–Whitney test). Abbreviations: IA = intra-arterial.

**Figure 7 ijms-23-12221-f007:**
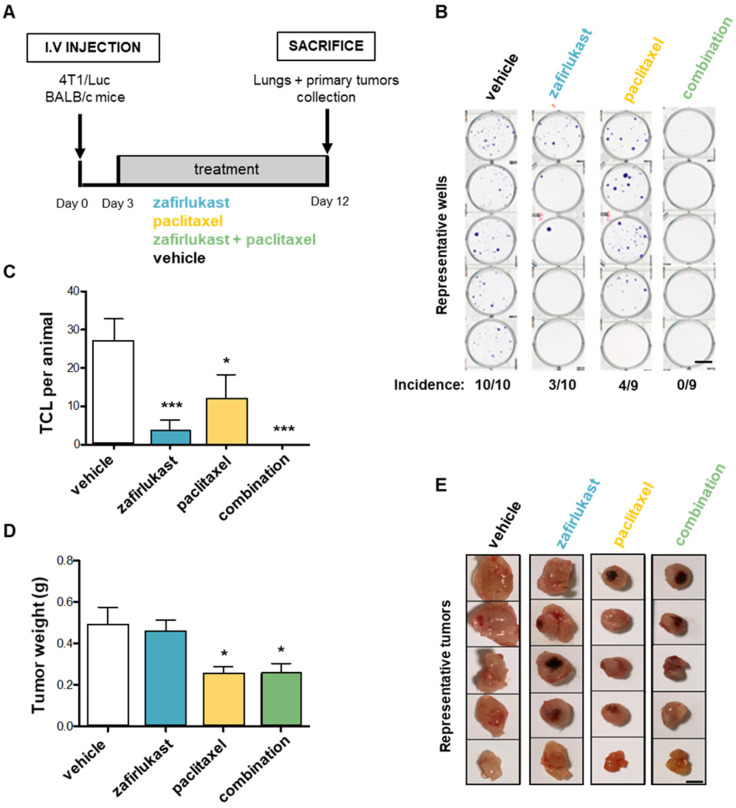
Blockade of CysLT1R with zafirlukast prevents spontaneous lung colonization by 4T1 cells without affecting orthotopic tumor growth in a syngeneic mouse model. (**A**) Murine 4T1 breast cancer cells were injected in the mammary gland of BALB/c mice that were treated with zafirlukast (0.4 mg/kg/day, i.p), paclitaxel (30 mg/kg/2 days), a combination of both drugs, or the vehicle. After 14 days, lungs and primary tumors were collected. Vehicle = physiological serum. (**B**) Lungs were minced and treated with collagenase and cultured for 14 days with 6-thioguanine (1 µg/mL). Tumor cells that colonized the lungs (TCL) were fixed and stained by crystal violet (scale: 100 µm). (**C**) TCL were quantified. (**D**) Mammary tumors were collected and weighed. (**E**) Primary tumors were classified by size. The data represent the mean ± SD (vehicle and zafirlukast: 10 mice; paclitaxel and combination: 9 mice). * *p* < 0.05; *** *p* < 0.001 vs. vehicle; Mann–Whitney test).

## Data Availability

All data generated or analyzed during the current study are included in this manuscript and its Appendix A.

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
