# Peer review of "Blockade of Platelet CysLT1R Receptor with Zafirlukast Counteracts Platelet Protumoral Action and Prevents Breast Cancer Metastasis to Bone and Lung"

_ijms, 2022, doi:10.3390/ijms232012221_

Round 1
Reviewer 1 Report
Dear Authors,
The manuscript presented here is well executed and written. I do not have any major concerns or suggestions.
I did notice some spelling mistake and missing text. For eg. Line 255 puromycin, Line 292, Figure number is missing. Please go through the manuscript to edit these mistakes.
Thanks
Author Response
The spelling mistake and missing text have been corrected.
Reviewer 2 Report
This manuscript presents a compelling anti-metastasis effector of zafirlukast on platelet-induced CTC survival. The author presents the molecular interactions and used a well-designed mouse model further demonstrates the inhibition effect of zafirlukast on breast cancer mets. The experiments were carefully designed with quantifiable results. Not only are direct anti-tumor effects mediated by blocking platelet cloaking through CysLT1R, but this paper also gives a comprehensive exploration of the interesting metabolic mechanism of CysLT in breast cancer lines.
The manuscript was well written with sufficient details for readers and is suitable for publication. Only two minor revision comments:
Minor:
1. What're the enzymes in the brackets mean in Figure2A? No corresponding legend describing these.
2. Figure 4. Is it possible to add a control of non-tumor line MDA-231 to further demonstrate the anti-tumor effect of zafirlukast is specific to tumor cells?
Author Response
- The legend of figure 2A and the corresponding text have been edited to precise that CysLT relate to LTC4, LTD4 and LTE4.
- MDA-231 are not non-tumor cell lines as they are also human triple negative breast carcinoma cells with a high metastasis profile. These cells are also known to induce tumor cell-induced platelet aggregation (TCIPA) a process that has been well described to contribute to platelet protumoral actions. Please see the reference Boucharaba et al, JCI 2004. In figure 4, we show that zafirlukast inhibits TCIPA. However, zafirlukast had no inhibitory effect on TRAP6-induced platelet aggregation as shown in supplementary data 1. These results indicate that zafirlukast can specifically inhibit TCIPA without affecting their physiological functions.
Reviewer 3 Report
Summary
The article “Blockade of Platelet CysLT1R Receptor with Zafirlukast Counteracts Platelet Protumoral Action and Prevents Breast Cancer Metastasis to Bone and Lung” by Saier et al. for the first time reports a novel pharmacologic strategy targeting blood platelet protumoral function in breast cancer. The authors showed that inhibition of CysLT1R using the leukotriene receptor antagonist zafirlukast successfully blocks the pathological metastatic tumor cell (MDA-B02)-platelet interaction and hence hampers the platelet-induced tumor progression and/or metastasis without abrogating normal platelet functions. The manuscript does an excellent job demonstrating both in vitro and in vivo that tumoral CysLT via platelet aggregation contributes to platelet-mediated breast cancer progression and/or metastasis and that treatment with zafirlukast effectively and specifically hinders metastatic progression while maintaining the natural hemostatic platelet function.
I believe that this paper is of high importance for the field of breast cancer research and should be taken as a basis for the development of novel anti-metastatic or even preventive therapies.
Moreover, the article is well-written, well-structured and appropriate methodology was applied. Data are clearly and concisely illustrated in the presented figures and the graphical abstract beautifully summarizes and highlights the relevance of the novel findings.
Only a few major issues given below should be addressed to further improve the article.
Major issues
1. Did the authors perform dose-finding experiments prior to in vitro and in vivo experiments in order to find appropriate treatment concentrations per drug? Please state in the methods part why you chose the given drug concentrations as done for zafirlukast in line360-362.
2. Drug screening and cell survival assay experiments were performed on only one breast cancer cell line MDA-B02-Luc. Why not also on parental MDA-213 and MDA-B02 cells?
3. Page 3, line 107: What is % Δ activity?
4. Page 3, figure 1C: Which data are shown? Mean ± SD? How often was the experiment reproduced (give n number)?
5. Page 4, line 112: What is 5-LOX? Give a brief introduction.
6. Page 5, figure 2A: Please show also negative and positive controls of the CysLT ELISA.
7. Page 5, figure 2B: The authors show via Western Blot that MDA-213, MDA-B02 ad MDA-B02-Luc cells do not express LTC4S protein. Have the authors investigated LTC4S gene expression via quantitative Real-time PCR for confirmation?
8. Page 7, line 191-195: Please explain what is detected by CD41 staining.
9. Page 10, figure 5A: Why did the authors choose the 5h time point? A later time point might show more significant results. Moreover, please show also images of cells with 10% FBS in comparison to cells with platelets. What is the control group, what does “NT” stand for? Clarify and define in the figure legend.
10. Page 10, figure 5C: Please show also fluorescence images of negative and positive controls.
11. Page 10, figure 5A-C: Surprisingly, control cells and non-treated/non-transfected cells do not show equal results. To avoid confusion, remove the control group from figure 5A-C and normalize results to non-treated/non-transfected cells.
12. Page 11, line 278-280: How did the authors know that the TCL cells were tumor cells? Please clarify in the text.
13. Page 15, line 389: The establishment of the MDA-B02-Luc cell line is not described in the given reference 46. Please add the corresponding reference.
14. Page 15, line 389: What is meant by “triple-negative”? Please clarify in the text.
15. Page 15, paragraph “Cell culture and Reagents”: Order information of the drugs montelukast and everolimus are missing. Please add.
16. Page 15, line 407: Please give the order information for arachidonic acid and calcium ionophore A23187.
17. Page 18, line 537-538: Please add project identification code and data of approval for the work with human blood platelets.
18. Page 16, line 424: Please add primer sequence or at least the order number. Which housekeeping gene was used for normalization? Add this information to this paragraph.
19. Page 16, line 428: Also here, please add the order number of the used siRNAs.
20. Page 17, line 492: What was used as placebo treatment? Please mention in the text.
21. Page 17, line 504: What is the purpose of treatment with 6-thioguanine? Please mention the order information in the text.
Minor issues
- Page 1, line 15: Plural of “metastasis” is “metastases”. Please correct.
- Page 2, line 34: “Metastases” instead of “metastasis”
- Page 2, line 40: “Intravasation” instead of “invasion”
- Page 4, line 118: Misspelling. “…it could be found…” instead of “…in could be found…”
- Page 4, line 134: “It is well known…” instead of “It is well know…”
- Page 4, line 140: Please add (AA) behind “…incubated in the presence of arachidonic acid…”.
- Page 15, line 376: “…hemostatic functions of platelets (Figure 8).” There is no figure 8 available. Do you mean the graphical abstract figure? Please correct.
- Page 16: line 421: Spelling error “perfored”. Please correct and replace by “performed”.
- Page 17, line 500: Remove “IV”.
Author Response
Major issues
- We performed a dose-response assay of zafirlukast on platelet-induced cancer cell survival (Figure 1C) indicating an absence of effect at 1µM and an IC50 of 10µM. Those two concentrations were thus used in the in vitro The doses used in the in vivo experiments were adapted from the doses used with humans. The paragraph 4.12 of the materials and methods section has been changed adding occordingly.
- The cell survival assay was based on the measurement of the luciferase activity of MDA-B02-Luc cells instead of a standard survival assay based on metabolic activity to avoid any influence of platelet activity on the measure. Thus the experiment could not be performed with parental MDA-231 and MDA-B02 cells.
- The legend of Figure 2B and the materials and methods have been corrected to define %∆ activity.
- The legend of Figure 1C has been edited to add the information requested.
- The text was modified to precise the function of 5-LOX.
- Figure 2A represents the results of an ELISA assay used for quantification of CysLT. A standard curve (r²=0.99) was calculated from a series of wells containing known amount of analyte on the same plate than the samples to validate the assay and the concentration indicated are corrected to take into account the background caused by the culture media according to the manufacturer instructions (Cayman #500390). These details have been included in paragraph 4.3 of the materials and methods section. We thus believe that Figure 2A should not be modified and comprises all the information needed.
- We did investigate LTC4S expression via RT qPCR and it was not detected. The results of the RT-qPCR have been added as supplemented data 3.
- We clarified in the text that platelets were detected using an anti-CD41 antibody.
- We choose a short 5h time point to avoid cell proliferation to interfere with the measure of cell migration. The images of cells with 10% FBS were added to figure 5A. The images and legend of figure 5A-C were modified to clarify what is the control group what NT stand for.
- Fluorescent images of positive and negative controls were added on figure 5C.
- We believe that control groups in figure 5A-C should be kept in order to show the effect of the addition of platelets on cancer cells migration and invasion compared to the controls. The results are not normalized because the results are presented as the percentage of wound closure and cell migration rate in µm per hour instead of relative numbers. We believe that relative numbers are less informative.
- The main text and materials and methods sections were modified to precise that TCL of 4T1 cells were rescued as they are naturally resistant to 6-Thioguanine but not normal bone marrow cells.
- The reference 46 has been replace by Peyruchaud et al. J Biol Chem
- « Triple-negative » correspond to a specific class of aggressive breast cancer cells. This information does not bring scientific added value to the text. So, the term “Triple-negative” has been withdrawn from the text to avoid confusion.
- Order information was added.
- Order information was added.
- The information was added in the institutional review board statement.
- Order numbers of the primers were added. The housekeeping gene was specified.
- The order numbers of the siRNA were added.
- The nature of the placebo treatment (physiological serum) has been specified in the text and legends.
- The text has been modified to specify that 6-thioguanine treatment enables to select 4T1 cancer cells as they are naturally resistant.
Minor issues
- Page 2, line 40 : the term « invasion » should not be changed for « intravasation » as the sentence is correct.
- All the others minors issues were corrected as suggested.